# Modified Coptic Cross Shaped Split-Ring Resonator Based Negative Permittivity Metamaterial for Quad Band Satellite Applications with High Effective Medium Ratio

**DOI:** 10.3390/ma15093389

**Published:** 2022-05-09

**Authors:** Md Bellal Hossain, Mohammad Rashed Iqbal Faruque, Mohammad Tariqul Islam, Mayeen Uddin Khandaker, Nissren Tamam, Abdelmoneim Sulieman

**Affiliations:** 1Space Science Centre (ANGKASA), Institute of Climate Change (IPI), Universiti Kebangsaan Malaysia, Bangi 43600, Selangor, Malaysia; p109529@siswa.ukm.edu.my; 2Department of Electrical, Electronic & Systems Engineering, Faculty of Engineering and Built Environment, Universiti Kebangsaan Malaysia, Bangi 43600, Selangor, Malaysia; tariqul@ukm.edu.my; 3Centre for Applied Physics and Radiation Technologies, School of Engineering and Technology, Sunway University, Bandar Sunway 47500, Selangor, Malaysia; mayeenk@sunway.edu.my; 4Department of Physics, College of Sciences, Princess Nourah bint Abdulrahman University, P.O. Box 84428, Riyadh 11671, Saudi Arabia; nmtamam@pnu.edu.sa; 5Department of Radiology and Medical Imaging, Prince Sattam bin Abdul Aziz University, P.O. Box 422, Al-Kharj 11942, Saudi Arabia; a.sulieman@psau.edu.sa

**Keywords:** negative permittivity, Coptic cross, satellite application

## Abstract

This research article describes a modified Coptic cross shaped split ring resonator (SRR) based metamaterial that exhibits a negative permittivity and refractive index with a permeability of nearly zero. The metamaterial unit cell consists of an SRR and modified Coptic cross shaped resonator providing quadruple resonance frequency at 2.02, 6.985, 9.985 and 14.425 GHz with the magnitude of −29.45, −25.44, −19.05, and −24.45 dB, respectively. The unit cell that was fabricated on a FR-4 substrate with a thickness of 1.6 mm has an electrical dimension of 0.074λ × 0.074λ; the wavelength (λ) is computed at the frequency of 2.02 GHz. The computer simulation technology (CST) microwave studio was employed to determine the scattering parameters and their effective medium properties, i.e., permittivity, permeability and refractive index, also calculated based on NRW (Nicolson–Ross–Weir) method through the implementation of MATLAB code. The frequency range of 2.02–2.995 GHz, 6.985–7.945 GHz, 9.985–10.6 GHz, and 14.425–15.445 GHz has been found for negative permittivity. An effective medium ratio (EMR) of 13.50 at 2.02 GHz shows that the proposed unit cell is compact and effective. The lumped component based equivalent circuit model is used to validate with simulation results. The proposed unit cell and its array were fabricated for experimental verification. The results show that the simulation result using CST and high-frequency structure simulator (HFSS) simulator, equivalent circuit model result using advanced design system (ADS) simulator and measurement results match each other better. Its near zero permeability, negative permittivity, negative refractive index, high EMR and simple unit cell design allow the proposed metamaterial to be used for S-, C-, X- and Ku-band satellite applications.

## 1. Introduction

The beginning of the metamaterial-based research using theoretical and experimental demonstration was considered by John Pendry and David R. Smith in 2000, after ignoring Veselago’s idea in a similar case for more than 30 years [1,2,3]. Metamaterials are outlined as a man-made periodic structure designed to alter the electromagnetic behavior of materials to achieve excellent properties not found in nature. Metamaterials can have different types, for example single negative (ε or µ), double negative (ε and µ), and a double positive. Metamaterials have become an advanced technology due to their ability to obscure their identities and manipulate light. With the crystallization of this new technology, further applications have originated in radar and satellite communications [4,5], sensing applications [6,7], microwave and terahertz absorber [8,9,10,11], superlens [12], cloaking devices [13,14], seismic protection [15], and specific absorption rate (SAR) detection [16,17].

A double negative (DNG) metamaterial was proposed in [18] based on the modified split H-shape that exhibits single resonance frequency of 6.296 GHz and an EMR of 6.7 fabricated on Rogers RO 3010 substrate material. In [19], an epsilon negative (ENG) metamaterial based on crossed line SRR printed on Rogers RT 5880 dielectric material was introduced that presents triple band resonance frequency covering C-, X- and Ku-band microwave applications. This metamaterial’s unit cell size is 10 × 10 mm^2^ and an EMR of 4.5. In [20], a double C shaped metamaterial for microwave applications was introduced that displays four resonance frequencies at 2.37, 6.12, 8.16, and 14.51 GHz. This metamaterial shows a good EMR of 10.55. A DNG metamaterial based on modified Z-shaped was presented in [21] that is suitable for wideband microwave applications, showing a dual resonance frequency at 7.32 GHz and 11.84 GHz with poor EMR of 4.09. In [22], a DNG metamaterial was proposed based on double C shape that exhibits multiband resonance covering S-, C-, and X- band with EMR of 7.44. A DNG metamaterial based on the double Z-shape was outlined in [23] that presents a triple resonance frequency at 7.3, 8.1, and 9.4 GHz. This metamaterial unit cell size is 8.5 × 8.5 mm^2^ showing an EMR of 4.8. The negative permittivity metamaterial based on double dumbbell shaped SRR was suggested in [24], that is fabricated on Rogers RT 6002 substrate material. This metamaterial exhibits five resonance frequency covering the S-, X-, and Ku-bands and also shows an excellent EMR of 11.51. The electrical dimension of this metamaterial is 0.087λ × 0.087λ, where λ is calculated at the first resonance frequency. Aziz et al. reported the single negative (SNG) metamaterial structure based on concentric ring crossed lines that employed Ku- and Ka-bands, fabricated on FR-4 substrate with an EMR of 4.44 [25]. In another study [26], a DNG metamaterial with size of 5.5 × 5.5 mm^2^ was introduced that covered C-band microwave applications. This metamaterial exhibits a moderate EMR of 8. Another study demonstrated a metamaterial cloaking operation based on the epsilon negative, in [27]. This cloaking operates in a C-band microwave region and hides the metal cylinder. A hexagonal split-ring resonator with a coupled aperture is proposed as a microwave absorber in [28]. This structure exhibits a dual resonance frequency at 3.56 GHz and 11.67 GHz that is fabricated on a FR-4 substrate covering C and X-band microwave applications. An EMR of this resonator is 8.4. A triple band double negative metamaterial based on the double L-shape was proposed in [29]. This resonator shows a quadruple resonance frequency at 7.69, 8.47, 13.14, and 12.04 GHz and also presents an EMR of 3.9. In addition, [30] presented a double H-shaped SRR based metamaterial that exhibits quadruple resonance for satellite application. The dimension of the unit cell is 9 × 9 mm^2^, showing an EMR of 10.75. In other research, a cross-coupled interlinked based negative permittivity metamaterial was introduced in [31]. This metamaterial displayed triple resonance at 4.15 GHz, 10.38 GHz, and 14.93 GHz that also shows an EMR of 8.03. This work was also validated through the equivalent circuit model and measurement results. The effect of the dielectric properties of biological materials on the propagation of electromagnetic waves was presented in [32]. This work also delineates the technique of implementing the negative permittivity-based metamaterial structure on a microwave applicator that is enhanced by the control of the electromagnetic field in microwave therapy. In recent years, great progress has been made in the field of SPP. Much SPP research has been conducted, especially for metamaterials and photonic crystals, as they can dramatically adjust the performance of SPPs [33]. The above literature review is explained through the explanation of metamaterial behavior with multiband characteristics. However, high EMR with multiband resonance exhibiting metamaterial structure design still needs to be improved.

Since metamaterials consider their EMR to be inadequate in many of the applications discussed above, this research outlines a modified Coptic cross shaped split ring resonator with negative permittivity metamaterial for quad band satellite applications. This metamaterial is fabricated on low cost FR-4 substrate material showing resonances at 2.02, 6.985, 9.985, and 14.425 GHz that cover S-, C-, X- and Ku-bands with an EMR of 13.50. The negative permittivity was found to be 2.02–2.995 GHz, 6.985–7.945 GHz, 9.985–10.6 GHz, and 14.425–15.445 GHz, respectively. The novelty of this work is that the proposed design can provide a unique shape with miniaturized size showing a dimension of 0.074λ ×0.074λ. The proposed metamaterial produced a quadruple resonance covering with four bands including S-, C-, X-, and Ku-band satellite applications. Moreover, it can provide wider bandwidth, i.e., 0.385 GHz (1.815–2.2 GHz), 0.92 GHz (6.41–7.33 GHz), 0.56 GHz (9.66–10.22), and 1.39 GHz (13.57–14.96 GHz). The developed modified Coptic cross shape exhibited a high effective medium ratio (EMR) of 13.50 that showed compactness, which is relatively high in this regard compared to the recently proposed metamaterials. The array results are almost like a unit cell due to low mutual coupling effect. A unique characteristic of this structure is that the resonance frequency can be tuned by modifying the sizes of the Coptic width. A widely used NRW method is used to achieve the effective medium properties. E-field, H-field, and surface current were also analyzed to understand the behavior of the metamaterial. In addition, this work describes the equivalent circuit modeling using ADS simulator and also analyzes the measurement results through the fabrication of unit cell and array of unit cells.

## 2. Metamaterial Structure Design and Simulation Geometry

The proposed metamaterial unit cell is represented on an inexpensive FR-4 dielectric material (PCBWay, Xiacheng, China) with a thickness of 1.6 mm, a physical size of 11 × 11 mm^2^, and electrical dimension of 0.074λ × 0.074λ. The FR-4 substrate has a dielectric constant of 4.4 with a dissipation factor of 0.02. The resonator of the unit cell is made of annealed copper with a thickness of 0.035 mm and a conductivity of 5.96 × 10^7^ S/m. The metamaterial resonator consists of a split ring resonator and modified Coptic cross shaped resonator, as indicated in Figure 1. The top view of the unit cell represents a resonating patch of the metamaterial, as shown in Figure 1a; on the other hand, the isometric view shows the annealed copper resonating patch with FR-4 substrate, as presented in Figure 1b. The length and width of the SRR are 10.4 mm and 1 mm, respectively. The split gap of the SRR is 0.2 mm, which is fixed by the trial and error method. The split gap of the modified Coptic cross is 0.4 mm. The design specification of the metamaterial is presented in Table 1. The time domain solver with hexahedral meshing system is used to derive the scattering parameters. The simulation scenario is displayed in Figure 2, where a transverse electromagnetic wave (TEM) is usually propagated through the proposed unit cell. As shown in Figure 2, PEC–PMC boundary geometry is used for simulation. The metamaterial unit cell and arrays of unit cells are placed between the two waveguide ports (one is the transmitting end, the other is the receiving end).

## 3. Effective Medium Parameters Extraction Method

The simulation approach of the proposed modified Coptic cross shape-based unit cell was executed using a time domain solver with hexahedral mesh in the frequency span of 1 to 16 GHz. The NRW technique was used to obtain the relative permittivity, relative permeability, refractive index, and impedance [34]. The equations were as follows to determine the effective parameters:(1)$$\;\;\;\;\;\;\;\;\;\;\;\varepsilon_{r}\sim \frac{2}{jk_{0}d}\times \frac{\left(1-V_{1}\right)}{\left(1+V_{1}\right)}\;$$
(2)$$\;\;\;\;\;\;\;\;\mu_{r}\sim \frac{2}{jk_{0}d}\times \frac{\left(1-V_{2}\right)}{\left(1+V_{2}\right)}$$
(3)$$\eta_{r}=\frac{2}{jk_{0}d}\times \sqrt{\left\{\frac{{\left(S_{21}-1\right)}^{2}-S_{11}{}^{2}}{{\left(S_{21}+1\right)}^{2}-S_{11}{}^{2}}\right\}}$$
(4)$$Z=\sqrt{\frac{{\left(1+S_{11}\right)}^{2}-S_{21}^{2}}{{\left(1-S_{11}\right)}^{2}-S_{21}^{2}}}$$
where, *V_1_* and *V_2_* are the composite terms, which is derived using transmission and reflection coefficient of the unit cell; $k_{0}=\frac{2\pi f}{c}$, *V_1_* = |*S*_21_| + |*S*_11_|, *V_2_* = |*S*_21_| − |*S*_11_|, and d is the substrate thickness; and *ε_r_*, *µ_r_*, *η_r_*, and *Z* are the relative permittivity, relative permeability, refractive index, and normalized impedance, respectively.

## 4. Design Procedure of the Metamaterial Unit Cell

The proposed design was chosen to analyze different unit cell design approaches based on a trial and error method, as shown in Figure 3, and the *S*_21_ and *S*_11_ plot for the different design approaches is shown in Figure 4a,b. Multiple resonance frequencies require more factors, i.e., the size and thickness of the substrate, width and split gap of the resonating patch, and mutual coupling of the resonator. The variation in capacitance and inductance of the unit cell is responsible for creating a multiple resonance frequency. The unit cell design was initiated with the insertion of a split ring resonator through the split gap of 0.2 mm. Three resonance frequencies were obtained from design 1, the resonance frequencies of reflection coefficients (*S*_11_) are 3.325, 9.05, and 11.5 GHz with the magnitude of −27.78, −9.88, and −11.04 dB, respectively, whilst the resonance frequencies of transmission coefficients (*S*_21_) are 2.05, 7.19, and 9.47 GHz with the magnitude of −29.42, −28.92, and −11.25 dB, respectively. In the next step, a Coptic shape is added to the previous design showing triple resonances. This inclusion improves the inductance value and thus creates a similar resonance with the previous design, but increased the magnitude by slightly changing the first and second resonance frequencies to a lower frequency range. The resonance frequencies of *S*_11_ are at 2.995, 7.39, and 15.97 GHz with the magnitude of −27.75, −12.15, and −3.57 dB; the *S*_21_ is at 2.02, 6.79, and 12.91 GHz with the magnitude of −28.48, −22.41, and −34 dB, respectively. According to design 3 of Figure 3, the same resonance frequency exists as the earlier design by inserting four cross shapes into design 2, because of only improving the inductive property. The magnitudes of the resonance frequencies of *S*_11_ are −27.52, −12.06, and −4.05 dB at 2.98, 7.39, and 15.8 GHz, respectively, whereas, the resonance frequencies of *S*_21_ are 2.02, 6.79, and 12.88 GHz with the magnitude of −28.97, −22.45, and −34 dB, respectively. The design is completed by inserting two split gaps in Coptic shape, as shown in Figure 3. The inserting split gap in design 3 creates additional resonance frequency due to generating a capacitance property. According to Figure 4, the resonance frequencies of *S*_11_ are 3.085, 8.095, 10.795, and 15.775 GHz with the magnitude of −27.03, −13.14, −9.66, and −4.91 dB; the resonance frequencies of *S*_21_ are 2.02, 6.985, 9.985, and 14.425 GHz with the magnitude of −29.45, −25.44, −19.05, and −24.45 dB, respectively. Consequently, the proposed modified Coptic cross shape-based metamaterial unit cell provides quadruple resonance frequency at 2.02, 6.985, 9.985, and 14.425 GHz covering S-, C-, X-, and Ku-band satellite applications.

## 5. Parametric Study

The performance of the metamaterial unit cell depends on several factors such as the split gap of the ring, the width of the metamaterial resonator, substrate and its thickness, and boundary condition of the simulation environment. These factors directly hit the variation of inductance and capacitance property, which results in a change in resonance frequency of the metamaterial. Numerous parametric studies have been executed to investigate the effects of the split gap of the SRR and Coptic cross resonator, the width of the Coptic cross resonator, effect of different conductor on metamaterial resonator, effect of substrate material, and boundary condition of the simulation geometry.

**Effect of split gap of SRR on *S*_21_ results**:

The *S*_21_ plot for the various split gap of SRR is introduced in Figure 5a. The split gap of the SRR initiates a capacitance variation that influences the resonance frequency of the metamaterial. The capacitance value decreases with the increasing split gap of the ring, which affects the resonance frequency to a higher value. The initial step introduces a split gap of 0.2 mm, exhibiting quadruple resonances with excellent magnitude. Then, the split gap was changed to 0.4 mm and 0.6 mm, showing the same response as the result of 0.2 mm split gap of SRR, but the resonance frequency moves to higher value. Finally, the no split gap is inserted inside the ring, which affects the resonance frequency, where the first resonance is moved to 4.14 GHz, and the second resonance frequency is almost diminished. It can be seen from Figure 5a that the resonance frequency for 0.2 mm split gap is better than other specified values. Finally, a 0.2 mm split gap is selected for the proposed design.


**Effect of split gap of Coptic shape resonator on *S*_21_ results:**


The *S*_21_ plot for the various split gaps of the Coptic shape resonator is shown in Figure 5b. The capacitance property depends on the split size of the resonator. The first design starts with a 0.2 mm split gap of the Coptic shape resonator showing quadruple resonance frequency with excellent magnitude. The split gap was uniformly increased by 0.2 mm. An increase in the value of the resonance frequency was also observed. Lastly, no split gap was inserted into the Coptic shape resonator that affects the resonance frequency, where the third resonance frequency was shifted from 10.3 to 12.895 GHz and the fourth resonance disappeared. As shown in Figure 5b, it can be realized that the resonance frequency with magnitude is better than the other specified values for a Coptic shape resonator split gap of 0.4 mm. Finally, a split gap of 0.4 mm was chosen for the proposed design.


**Effect of Coptic shape resonator width on *S*_21_ results:**


The *S*_21_ plot for the different widths of the Coptic shape resonator is outlined in Figure 5c. As the Coptic shape resonator’s width increases, the overall inductance property decreases and the resonance frequency increases, because the resonance frequency is inversely proportional to the inductance. The width of the Coptic shape resonator was taken in four sizes, such as 0.4, 0.6, 0.8, and 1 mm. The effect of width analysis found that the first and second resonance frequencies were almost the same for different widths, but the third and fourth resonance frequencies increased slightly. In this analysis, the width was set to 0.6 mm for the proposed design.


**Effect of different resonating conductor on *S*_21_ results:**


The effect of different resonating conductors on the resonance frequency of *S*_21_ is described in Figure 5d. The different metals such as gold, nickel, platinum, tantalum, and copper are used to test the performance of the metamaterial. First, copper is used as a patch to indicate the quadruple resonance frequency with excellent bandwidth. The effect of a resonating conductor (gold, platinum, tantalum) indicated that the response of *S*_21_ was the same as copper, whereas nickel used as a patch showed the same response with lower magnitude. Since gold, platinum, and tantalum are very expensive metals, annealed copper was used as a resonating conductor for the proposed design.


**Effect of substrate material on *S*_21_ response:**


Three different substrate materials were used for substrate selection. The three substrates are FR-4 (lossy and loss-free), Rogers RT-5880 (lossy), and Rogers RT-6002 (lossy). All substrate properties are different. In the first analysis, the dielectric constant, dielectric loss, and thickness (mm) of FR-4 (lossy) are 4.3, 0.025, and 1.6, respectively. Figure 6a shows that this material has demonstrated quadruple resonance frequency with the magnitudes of −29.45, −25.44, −19.05, and −24.45 dB; on the other hand, FR-4 (loss-free) material exhibits the same resonance with the magnitudes of −39.35, −44.55, −38.94, and −49.02 dB. The dielectric constant, dielectric loss, and thickness (mm) of Rogers RT-5880 (lossy) and RT-6002 (lossy) are 2.2, 0.0009, and 1.575 and 2.94, 0.0012 and 1.524, respectively. As shown in Figure 6a, both substrate materials exhibit triple resonance frequency with good magnitude. Compared to the FR-4 substrate, the resonance frequency of both Rogers materials moves to a higher value. However, the performance of the FR-4 substrate material is better than the other substrate materials. Therefore, the FR-4 substrate material was selected solely because it was consistent with the purpose of this study.


**Effect of substrate length on *S*_21_ response:**


The design starts with an initial selection of basic dimensions and considers the same general subwavelength criteria as λ/10. The *S*_21_ plot for different substrate lengths is exhibited in Figure 6b. Since the substrate length and resonance frequency are inversely proportional to each other, we examined the performance of four different unit cell substrate length, such as 14, 13, 12, and 11 mm, in order to obtain the lowest resonance frequency in the lower dimension as the higher EMR. For the effect of substrate length on *S*_21_ response, it was found that first and second resonance frequency shifted to higher value but third and fourth resonance frequency decreased slightly. The unit cell with four different substrate lengths exhibits the same resonance frequency. Finally, the proposed metamaterial resonator with a substrate length of 11 mm on each side is selected to achieve quadruple resonance frequency covering for S-, C-, X-, and Ku-band.


**Effect of different boundary condition on *S*_21_ response:**


The aim of this study is to elucidate the effects of different boundary conditions on modified Coptic cross shape-based metamaterial in the frequency range of 1 to 16 GHz. Five different boundary conditions, perfect electric conductor (PEC)– perfect magnetic conductor (PMC), PMC–PEC, PEC–PEC, PMC–PMC, and Unit Cell, are used to describe the metamaterial performance. The proposed metamaterial is used to demonstrate that the use of different boundary conditions can lead to completely different electromagnetic responses, as shown in Figure 6c. In the first study, PEC–PMC (E-field in *x*-axis and H-field in *y*-axis) boundary conditions are used in modified Coptic cross shape-based metamaterial; there are quadruple resonance frequencies in a given frequency range, representing 2.02, 6.985, 9.985, and 14.425 GHz. In the second study, PMC–PEC (H-field in *x*-axis and E-field in *y*-axis) boundary conditions are utilized in the proposed metamaterial exhibiting five resonance frequencies with smaller magnitude compared to PEC–PMC conditions. When PEC–PEC (E-field in *x*-axis and *y*-axis) conditions are employed in the proposed metamaterial, three non-uniform resonance frequencies arise in a given frequency range. In the orientation of PMC–PMC (H-field in *x*-axis and *y*-axis), conditions apply in both x and y directions. As seen from Figure 6c, these boundary conditions have a completely different response from the other boundary conditions. Lastly, unit cell boundary conditions (unit cells repeat periodically in *x*-axis and *y*-axis) are used to see the response of metamaterial. This boundary condition shows only one resonance peak. However, only the PEC–PMC (E-field in *x*-axis and H-field in *y*-axis) boundary condition was preserved because it fulfilled the purpose of this study.

## 6. Equivalent Circuit of the Proposed Unit Cell

The equivalent circuit model of the proposed unit cell is made up of passive components, such as inductor (*L*) due to metallic resonator patch, and capacitor (*C*) due to split gap of the ring resonator. The split gap of the ring with metallic resonator creates LC resonance circuits that supply the resonance frequency at a certain frequency, which is changed by the variation in the split gap and metallic strip of the metamaterial resonator [35].
(5)$$f=\frac{1}{2\pi \sqrt{LC}}\;$$

The overall inductance was computed according to the principle of the transmission line [36]:(6)$$L=0.01\times \mu_{0}\left\{\frac{2{\left(d+g+h\right)}^{2}}{{\left(2w+g+h\right)}^{2}}+\frac{\sqrt{{\left(2w+g+h\right)}^{2}+l^{2}}}{\left(d+g+h\right)}\right\}t$$

Furthermore, the overall capacitance was calculated using:(7)$$C=\epsilon_{0}\left[\frac{2w+g+h}{2\pi {\left(d+h\right)}^{2}}\ln\left\{\frac{2\left(d+g+h\right)}{\left(a-l\right)}\right\}\right]t\;$$
where *ε_0_* = 8.854 × 10^−12^ F/m, *μ_0_* = 4π × 10^−7^ H/m, *w* = metallic resonating patch width, *h* = substrate thickness, t = metallic resonating patch thickness, and *l* = length.

The optimized equivalent circuit of the proposed metamaterial unit cell using the passive element (*L*, *C*) responsible for generating the quadruple resonance frequency is shown in Figure 7a. The component values are adjusted using an ADS simulator. By adjusting the component values, the *S*_21_ results are computed so that the ADS results are very similar to the CST results. A comparison of *S*_21_ is shown in Figure 7b. It was observed that the equivalent circuit results were expressed identically to those observed in unit cell results using CST in case of first to third resonance frequency. The metamaterial resonator consists of a split ring resonator and two split based Coptic cross shape resonator. The split ring resonator can be represented by L1, C1, L2, and C2, which is responsible for the creation of resonance frequency of 2.02 and 6.985 GHz. The Coptic shape with two split rings contributes to the creation of two resonance frequencies at 9.985 and 14.425 GHz through the representative L3, C3, L4, and C4 component of proper tuning. In the equivalent circuit model, L5 and L6 boost the magnitude of resonances representing the equivalent of the four cross shaped resonators. C5 represents the coupling capacitor.

## 7. Results and Discussion

The numerical simulation of the modified Coptic cross shape-based metamaterial unit cell was executed using a Finite Integration Technique (FIT) based CST simulator. The scattering parameters (*S*_11_ and *S*_21_) graph of the proposed unit cell is delineated in Figure 8a. The *S*_11_ results of the proposed unit cell provide quadruple resonances with frequencies of 3.085, 8.095, 10.795, and 15.775 GHz with an amplitude of −27.03, −13.14, −9.66, and −4.91 dB; on the other hand, the *S*_21_ results exhibit four resonance frequencies at 2.02, 6.985, 9.985, and 14.425 GHz with the magnitude of −29.45, −25.44, −19.05, and −24.45 dB, respectively. The unit cell provides excellent bandwidth for *S*_21_ results in the range of 1.815 to 2.202 GHz, 6.415 to 7.335 GHz, 9.658 to 10.208 GHz, and 13.575 to 14.959 GHz. To verify the results, numerical simulation was also performed using a Finite Element Method (FEM) based HFSS simulator. The scattering parameters’ results of the proposed unit cell are outlined in Figure 8b. The *S*_11_ results of the proposed unit cell contribute four resonances with frequencies of 3.09, 8, 10.76, and 15.46 GHz with an amplitude of −27.04, −13.30, −9.72, and −5.56 dB; the *S*_21_ results show four resonance frequencies at 2.05, 6.97, 9.98, and 14.215 GHz with the magnitude of −30.65, −27.26, −20.55, and −25.20 dB, respectively. From the Figure 8b, it can be seen that the proposed unit cell exhibited quadruple resonances in both simulators with slight fluctuations.

To further validate the results, the measurement approach through the fabrication of the unit cell was performed using the Agilent N5227 PNA Microwave Network Analyzer. The measurement prototype was set up between a pair of waveguides. There are four pairs of waveguides used in this measurement process, including 510WCAS (1.45–2.20 GHz), 137WCAS (5.85–8.20 GHz), 90WCAS (8.20–12.4 GHz), and 75WCAS (10–15 GHz). The prototype and measurement setup of the proposed unit cell and array are shown in Figure 9 a–c. The measured resonance frequencies of *S*_21_ are 1.77, 7.181, 9.787, and 14.425 GHz with an amplitude of −11.89, −28.45, −20.44, and −33.73 dB. Figure 9d shows that the measured transmission coefficient (*S*_21_) results (second, third, and fourth) are slightly inconsistent with the simulation results due to vector network analyzer (VNA) calibration errors, waveguide coupling effects, and cable losses. The first resonance frequency of measured results is more varied with simulated results. The proposed metamaterial structure has explored the tunability properties. Smaller Coptic resonator width sizes result in lower annealed copper inclusion, resulting in lower ohmic losses. Meanwhile, a large Coptic width size has several advantages in the attempt to obtain an infinite medium at higher frequency. This indicates that by changing the Coptic width, the metamaterial structure can be tailored for a variety of applications of interest. The influence of the Coptic width of the metamaterial on the transmission coefficient has already been investigated in Figure 5c. The Coptic width size ranges from 0.4 mm to 1 mm. The resonance frequencies shift higher as the Coptic width size increases. A unique characteristic of this structure is that the resonance frequency can be tuned by modifying the sizes of the Coptic width. Selecting the resonance frequency through a tuning process based on the size of the Coptic width makes it more suitable for satellite applications.

The effective medium parameters (ε, µ, η, and Z) of modified Coptic cross shape-based negative permittivity metamaterial are extracted from the real and imaginary value of *S*_11_ and *S*_21_ using the NRW method, considered by Equations (1)–(4). The NRW-based MATLAB code is used to achieve the real and imaginary value of effective medium properties, as shown in Figure 10. Figure 10a shows a plot of relative permittivity versus frequency with negative permittivity (real) in the frequency range of 2.02 to 2.995, 6.985 to 7.945, 9.985 to 10.6, and 14.425 to 15.475. The imaginary values of effective permittivity are positive, and thus fulfill the criteria for metamaterial behavior. The effective permeability (real) values were near to zero at each resonance, with an amplitude of 0.048, 0.084, 0.16, and 0.12, respectively, as indicated in Figure 10b. The refractive index versus frequency plot is shown in Figure 10c, which shows the negative refractive index (real) in the 2.065 to 2.86 GHz, 7.09 to 7.765 GHz, 10.18 to 10.315, and 14.725 to15.04 GHz frequency ranges. On the other hand, the imaginary value of the refractive index exhibits a positive scenario in the same frequency range that reduces the incident wave in the structure. The impedance versus frequency plot is outlined in Figure 10d. It shows positive impedance (real) values over the entire frequency range. Impedance is observed close to zero, such as 0.006, 0.016, 0.04, and 0.029, at the resonance frequencies of 2.02, 6.985, 9.985, and 14.425 GHz, respectively, that represent the proposed metamaterial performing as a passive medium. The proposed metamaterial structure is effective or homogenized due to fulfilling the criteria for metamaterial behavior. The impedance (real), permittivity (imaginary), permeability (imaginary), and refractive index (imaginary) of the proposed metamaterial structure exhibits positive value, as shown in Figure 10. Moreover, the proposed metamaterial structure is used in S-, C-, X-, and Ku-band satellite applications. The C-band is used in satellite communication between the ground station and satellite. This band is also used in weather radar systems, terrestrial microwave links, and Wi-Fi devices. The S- and X-band are used in the weather radar system and communication satellite systems. Meanwhile, the Ku-band is commonly employed in satellite TV networks. Therefore, the metamaterial unit cell design outlined in this work can be effectively used in satellite. However, using this metamaterial structure offers various microwave products in S-, C-, X- and Ku-bands for satellite applications. This proposed metamaterial design can be used to make a variety of devices including antennas, bandpass filters, power dividers, and couplers. This metamaterial structure essentially enhances the performance of the satellite antenna.

## 8. Surface Current, Magnetic Field, and Electric Field Analysis

In a metallic resonator, the surface current delineates the actual current generated by the time varying electromagnetic field. This actual current produces a magnetic field, so the changing magnetic field produces an electromotive force. Maxwell and conventional law have proposed several equations to elucidate the relationship between surface current, E-field, and H-field through the following equations [37]:(8)$$\nabla \times H=J+\frac{\partial D}{\partial t}$$
(9)$$\nabla \times E=-\frac{\partial B}{\partial t}$$
where
(10)$$\nabla =\left[\frac{\partial }{\partial x},\frac{\partial }{\partial y}\;,\;\frac{\partial }{\partial z}\;\right]$$
(11)D(t)=ε(t)∗E(t)
(12)B(t)=μ(t)∗H(t)
where *E*, *H*, *D*, *B*, *ɛ*, *μ*, and *J* are *E*-field intensities, H-field intensities, electric flux densities, magnetic flux densities, electric permittivity, magnetic permeability, and time-varying electric current density in a medium, respectively.

Figure 11 displays the surface current, H-field, and E-field distribution of the proposed unit cell at the frequencies of 2.02, 6.985, 9.985, and 14.425 GHz.

At a frequency of 2.02 GHz, the outer SRR provides a low impedance path, thereby creating a strong surface current, as shown in Figure 11(ai). In Figure 11(bi), a good understanding was observed between surface current and H-field distribution, which was confirmed by Ampere’s law ($\int B.dl=\mu_{0}I$). As the current increases, the intensity of the H-field also increases. Strong surface current creates a higher H-field in same place. According to Maxwell’s law, the greater the value of the H-field in SRR, the greater value of the E-field on the other side of the SRR due to capacitive effect is shown in Figure 11(ci).

At the resonance frequency of 6.985 GHz, the outer SRR produces a little bit of current. In the other portion, the surface current was evenly distributed, as shown in Figure 11(aii). The Coptic shape resonator also contributed less to the surface current flow due to increasing the capacitive response for split gap. On the other hand, the SRR shows a small strength of H-field; the Coptic cross shape resonator has almost no current due to the split gap and the H-field approaches zero, as shown in Figure 11(bii). Meanwhile, the split gap of the SRR and Coptic shape resonator create an additional electric field due to forming a capacitor, as indicated in Figure 11(cii).

At a frequency of 9.985 GHz, the entire unit cell resonator, except the split gap of the SRR and Coptic shape, four cross shaped resonators, contributed most of the surface current, as indicated in Figure 11(aiii). This higher surface current creates a strong H-field through the same place of the resonator, as shown in Figure 11(biii). As shown in Figure 11(ciii), an increased E-field is observed throughout the structure at this resonance frequency, except in the resonator where the H-field is larger.

At the resonance frequency of 14.425 GHz, lower current densities were recorded across the unit cell as the impedance increased with higher frequencies, as shown in Figure 11(aiv). From the Figure 11(biv), it can be seen that the concentration of H-field remains the same where the surface current is higher. Meanwhile, the E-field strength is high where the H-field is low as a result of creating a mutual capacitance in a metamaterial unit cell, as shown in Figure 11(civ). The scale and axis views are shown in Figure 11v for the distribution of surface current, H-field, and E-field.

## 9. Array Metamaterial Results

The metamaterial unit cell does not act alone to exhibit the correct exotic electromagnetic properties; some series of array designs were simulated. The array performance was analyzed by choosing a few array types, such as 1 × 2 array (11 × 22 mm^2^), 2 × 2 array (22 × 22 mm^2^), and 4 × 4 array (44 × 44 mm^2^). This selected array was numerically simulated, using CST simulator as the same method of unit cell. Figure 12 shows the results of the scattering parameters for selected types of array. From Figure 12a, it can be seen that the *S*_11_ result of all arrays shows the same response only for the first and second resonance frequency. The *S*_11_ results of the unit cell and 1 × 2 array reveal the same response. The third and fourth resonance frequency of the 2 × 2 array and 4 × 4 array are shifted to the lower frequency due to the mutual coupling effect. Figure 12(b) presents the *S*_21_ results of changing at each resonance frequency for selected types of array. The *S*_21_ results of the unit cell and 1 × 2 arrays exhibit the same resonance frequency at 2.02, 6.985, 9.985, and 14.425 GHz with the magnitude of −29.45, −25.44, −19.05, and −24.45 dB, respectively. The third and fourth resonance frequencies shift lower for the 2 × 2 array and 4 × 4 array compared to the unit cell. The first and second resonance frequency for all the arrays exhibits the same response. The mutual coupling effect is responsible for this discrepancy between the resonance frequencies.

## 10. Effective Medium Ratio and Comparison with Recent Study

The effective medium ratio (EMR) is the fundamental aspect of metamaterial related applications. EMR stands for the smallness and usefulness of metamaterials. Many devices operate at low resonant frequencies, but the small size of the metamaterial makes it difficult to achieve low resonant frequencies. A high EMR represents the integrity and achievement of metamaterial design standards. If the EMR is less than 4, the subwavelength criteria of the metamaterial are not met. The operating frequency and size of metamaterials are inversely related to each other. Therefore, the EMR should be considered to design the metamaterial. EMR is the ratio of wavelength to unit cell length and is expressed by Equation (13).
(13)$$\mathrm{EMR}=\frac{\mathrm{Wavelength},\;\lambda }{\mathrm{Unit}\;\mathrm{cell}\;\mathrm{length},\;L}\;$$

The modified Coptic cross shape-based metamaterial unit cell exhibits an EMR of 13.50, 3.90, 2.73, and 1.89 at the frequency of 2.02, 6.985, 9.985, and 14.425 GHz. An EMR of 13.50 at 2.02 GHz shows that the proposed unit cell is compact and effective. The main advantages of high EMR are: (1) improved characteristic uniformity, (2) shorter electrical dimension without fabrication limitations, and (3) reduced transmission coupling effects. The unit cell reveals an electrical dimension of 0.074λ × 0.074λ; λ is calculated at the frequency of 2.02 GHz which fulfills the criteria of high EMR condition.

The performance of the designed metamaterial is compared with recently published work considering unit cell size (physical and electrical), a range of resonance frequencies with covered bands, and the effective medium ratio of the metamaterial. The comparison is shown in Table 2. The EMR is an important aspect for metamaterial. The proposed metamaterial exhibits an excellent EMR of 13.50, compared to other works listed in Table 2. The table shows that Refs. [24,30,31,38,39] have the highest electrical dimensions and lower EMRs compared to the proposed work; the proposed work is, therefore, the most effective. The proposed metamaterial contains moderate size and quadruple resonance with higher EMR, which makes this design more fulfilling than others shown in the comparison Table 2.

## 11. Conclusions

In this article, numerical and measurement representation of a modified Coptic cross shape-based metamaterial is presented in which the unit cell and array structure was designed, simulated, and investigated using CST and HFSS simulator. The proposed unit is fabricated on a FR-4 substrate with an electrical dimension of 0.074λ × 0.074λ, and λ is calculated at the frequency of 2.02 GHz. The simulated resonance frequencies using CST and HFSS are 2.02, 6.985, 9.985, and 14.425 GHz; 2.05, 6.97, 9.98, and 14.215 GHz, respectively. The measured resonance frequencies using Agilent N5227 PNA Microwave Network Analyzer are 1.77, 7.181, 9.787, and 14.425 GHz. The simulated results cover the S-, C-, X-, and Ku-band; additionally, measured results incorporate the L-, C-, X-, and Ku-band. In addition, the effective permittivity shows as negative at all resonance frequencies, whilst effective permeability exhibits near zero characteristics that treat the metamaterial as an SNG. Comparing simulation and measurement results shows strong agreement. The equivalent circuit model is sketched in the ADS simulator and the response of *S*_21_ agrees very well with the CST and HFSS simulation results. This study was also rigorously performed using surface current, H-field, and E-field analysis, and the overall performance is superior and better than other work described in this article. The inexpensive FR-4 substrate and small electrical size provide the desired resonance frequency and exhibit a high EMR of 13.50. Therefore, the proposed modified Coptic cross shape-based metamaterial can be widely used for S-, C-, X-, and Ku-band satellite applications. S-band is employed for Mobile Satellite Service (MSS) networks, C-band can be used in commercial telecommunications via satellites, X-band satellite communication is utilized by military forces, and Ku-band can also be applied to satellite TV networks and satellite communication systems.

## Figures and Tables

**Figure 1 materials-15-03389-f001:**
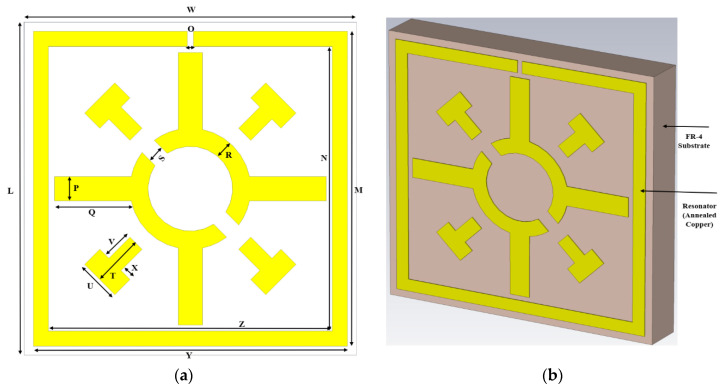
Different views of the unit cell: (**a**) Top view, (**b**) Isometric view.

**Figure 2 materials-15-03389-f002:**
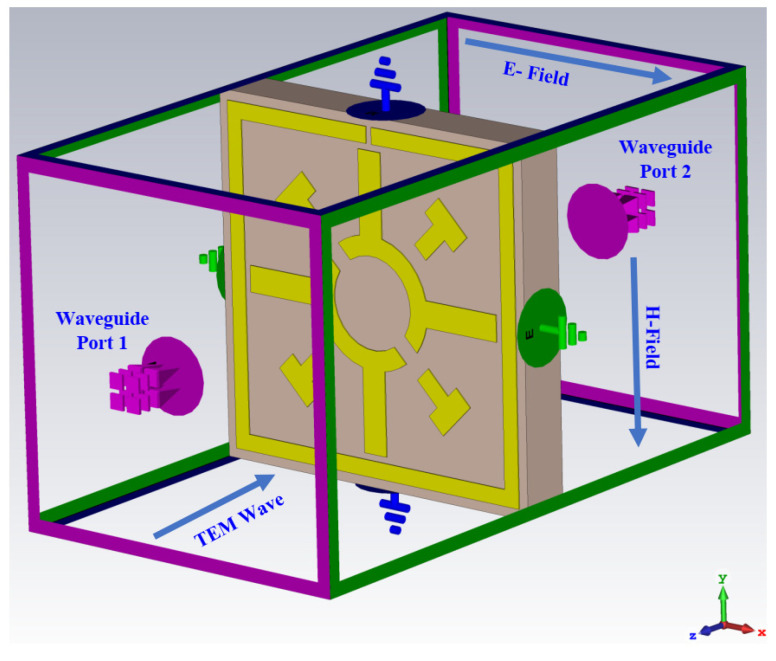
Simulation geometry of the proposed unit cell.

**Figure 3 materials-15-03389-f003:**
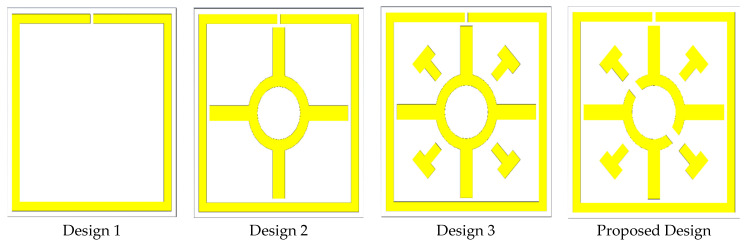
Different design approach towards finalizing the desired design.

**Figure 4 materials-15-03389-f004:**
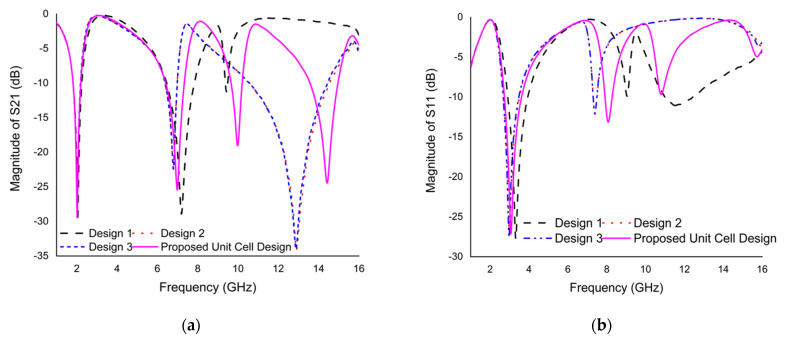
(**a**) The *S*_21_ plot on different design approaches. (**b**) The *S*_11_ plot on different design approaches.

**Figure 5 materials-15-03389-f005:**
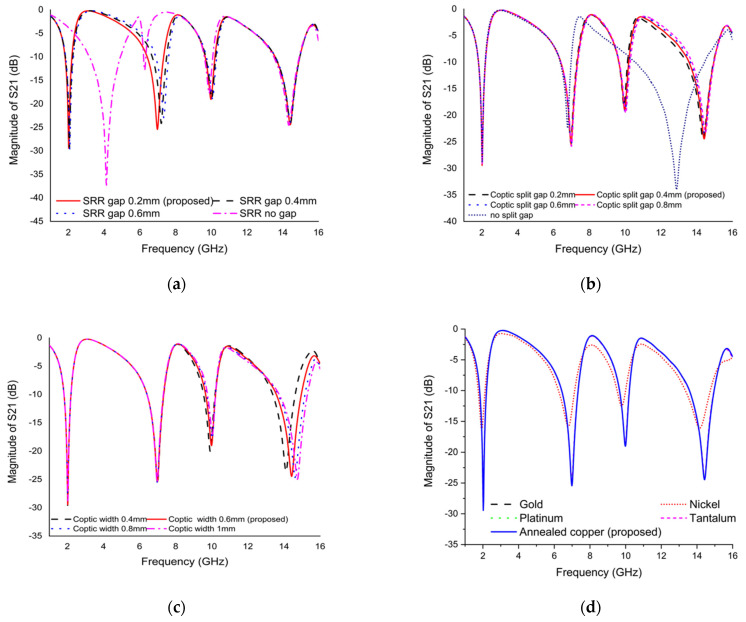
The *S*_21_ plot for (**a**) changing of split gap of SRR, (**b**) changing of split gap of Coptic cross resonator, (**c**) varying the width of the Coptic cross resonator, and (**d**) effect of different conductors used in resonator.

**Figure 6 materials-15-03389-f006:**
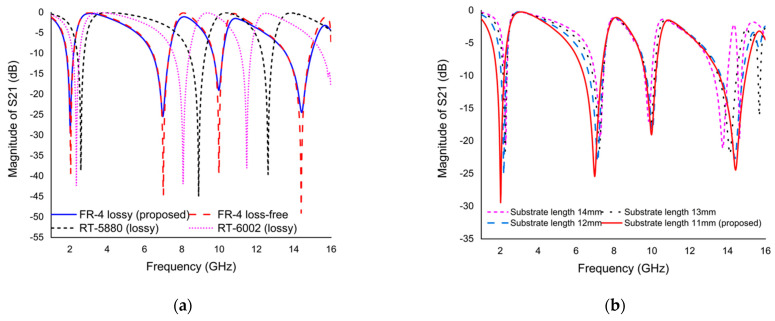
The S21 plot for (**a**) effect of substrate material, (**b**) effect of substrate length, and (**c**) the effect of different boundary conditions.

**Figure 7 materials-15-03389-f007:**
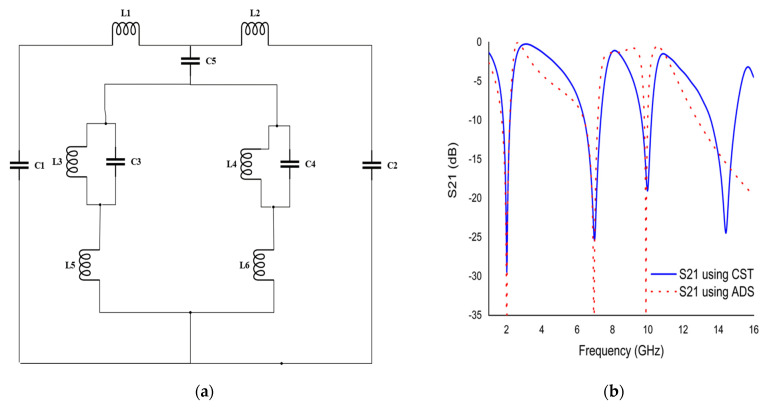
(**a**) Equivalent circuit model of unit cell. (**b**) S21 plot using CST and ADS simulator.

**Figure 8 materials-15-03389-f008:**
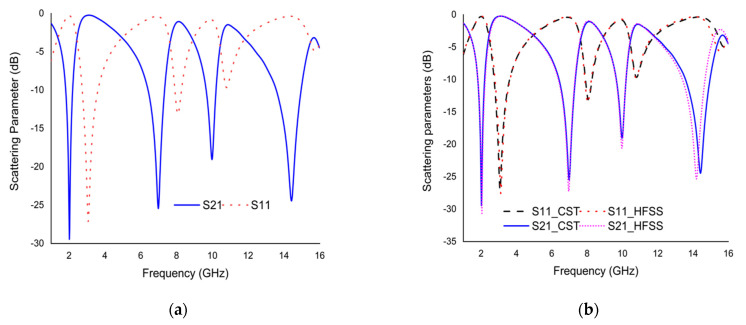
The graph for (**a**) *S*_11_ vs. *S*_21_ using CST simulator, (**b**) *S*_11_ vs. *S*_21_ using CST and HFSS simulator.

**Figure 9 materials-15-03389-f009:**
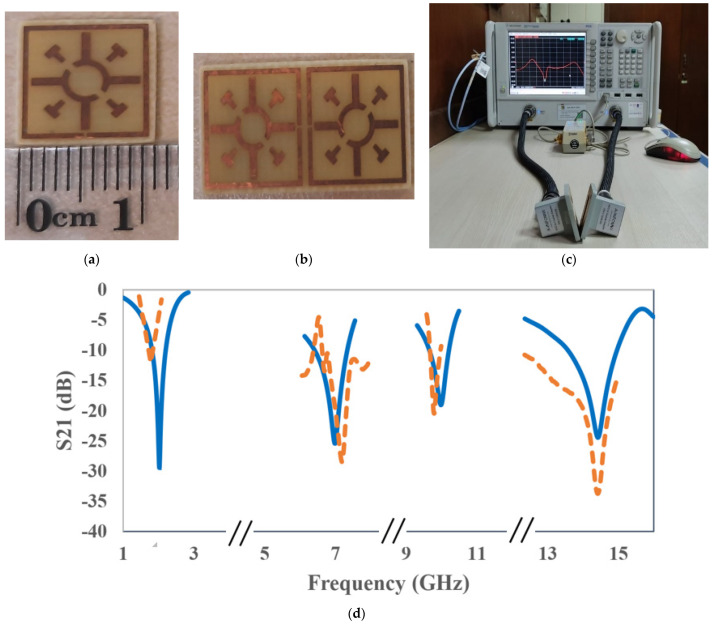
Graphical view of the (**a**) fabricated prototype of the unit cell, (**b**) 1 × 2 array, (**c**) measurement setup, and (**d**) *S*_21_ using simulated and measured results.

**Figure 10 materials-15-03389-f010:**
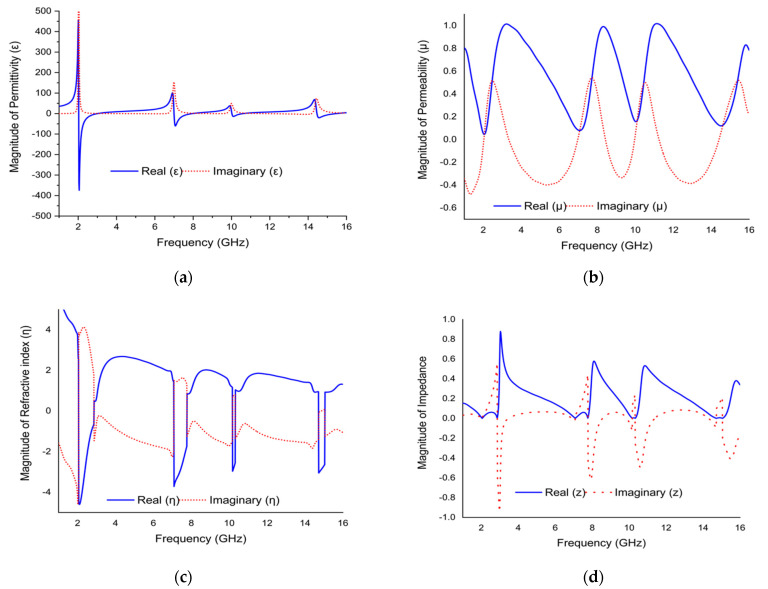
The graph of (**a**) permittivity vs. frequency, (**b**) permeability vs. frequency, (**c**) refractive index vs. frequency, and (**d**) impedance vs. frequency.

**Figure 11 materials-15-03389-f011:**
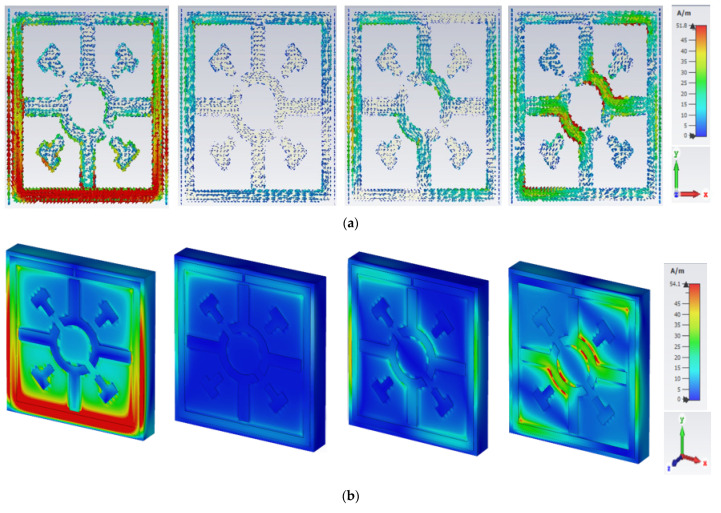
E-field, H-field, and surface current distributions of the Coptic cross shape-based metamaterial unit cell. (**a**) Surface current at top plane, (**b**) magnetic Field (H-field), (**c**) electric Field (E-field), (**i**) 2.02 GHz, (**ii**) 6.985 GHz, (**iii**) 9.985 GHz, (**iv**) 14.425 GHZ, and (**v**) scale and axis.

**Figure 12 materials-15-03389-f012:**
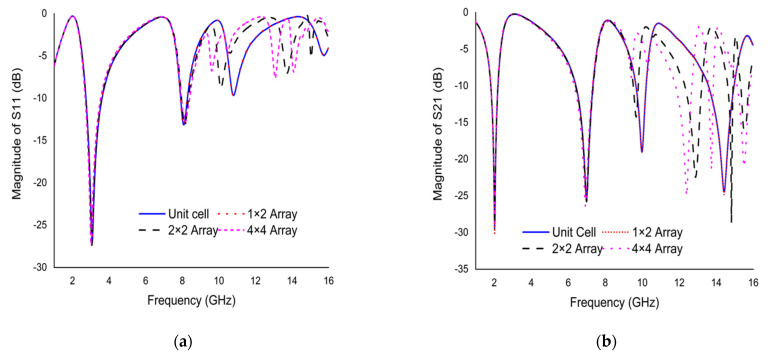
Scattering parameters plot using different types of array (**a**) *S*_11_ (**b**) *S*_21_.

**Table 1 materials-15-03389-t001:** The design specification of modified Coptic cross shape-based metamaterial.

Parameter	Dimension (mm)	Parameter	Dimension (mm)	Parameter	Dimension (mm)
W	11	S	0.4	V	0.8
L	11	T	1.2	X	0.3
P	0.8	U	1.0	Y	10.4
Q	2.7	M	10.4	Z	9.4
R	0.6	N	9.4	O	0.2

**Table 2 materials-15-03389-t002:** Comparison with recent published metamaterial work.

References	Published Year	Dimension (Physical and Electrical)	Resonance Frequencies (GHz)	Frequency Band	EMR
[24]	2021	9 × 9 mm^2^ 0.087λ × 0.087λ	2.896, 8.11, 9.76, 12.48, 13.49	S, X, Ku	11.51
[30]	2020	9 × 9 mm^2^ 0.094λ × 0.094λ	3.1, 10.1, 11.8, 12.44	S, C, X	13.16
[31]	2020	9 × 9 mm^2^ 0.124λ × 0.124λ	4.15, 10.38, 14.93	C, X, Ku	8.03
[38]	2021	10 × 10 mm^2^ 0.14λ × 0.14λ	4.20, 10.14, 13.15, 17.1	C, X, Ku	7.14
[39]	2021	8 × 8 mm^2^ 0.076λ × 0.076λ	2.86, 5, 8.30	S, C, X	13.1
This work	2022	11 × 11 mm^2^ 0.074λ × 0.074λ	2.02, 6.985, 9.985, 14.425	S, C, X, Ku	13.50

## Data Availability

All the data are available within the manuscript.

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
