# Peer review of "Modified Coptic Cross Shaped Split-Ring Resonator Based Negative Permittivity Metamaterial for Quad Band Satellite Applications with High Effective Medium Ratio"

_materials, 2022, doi:10.3390/ma15093389_

Round 1

Reviewer 1 Report

The paper describes a metamaterial that exhibits negative permittivity at four different frequencies. The proposed metamaterial is a modified Coptic cross-shaped split-ring resonator with a high effective medium ratio and negative permittivity. In general, a split-ring resonator produces a magnetic response, and a periodic array of such resonators can achieve negative permeability. The investigation in this paper does not align with the SRR theory, as the SRR is responsible for negative permittivity rather than negative permeability in this paper. The authors should provide a compelling justification for such behavior.
Furthermore, at the frequencies of interest, the proposed structure has a permeability of less than one. As a result, the material cannot be referred to as a negative permittivity material. The author should go over the analysis again and use the extraction formulas carefully, as there are many in the literature with different assumptions.

The paper is disorganized and poorly written.

Author Response

As attached.

Reviewer 2 Report

The manuscript by Hossain et al. demonstrated a novel design of Coptic cross shape split-ring resonator for quad band satellite applications. This is a comprehensive study of such system by combining both design, simulation, and experiments. I would suggest publishing this manuscript after a minor revision. Here are my comments on it:

  • The authors stated that the novelty of this study is the unique design shape showing quadruple resonance and quite high EMR. However, I think this might not be sufficient. The authors are advised to give a more detailed explanation of this by comparing this with literature (especially these adopting Coptic cross shaped ones).
  • For the designed pattern, it is a bit confusing. What is the reasoning for design from pattern 2 to pattern 3? Figure 4 showed the resonances are almost identical. Why are the four cross shape objects kept in pattern 4? Are they necessary? Could the pattern be simplified?
  • Section 5 is a bit to repeating in discussing different parameters. I would like to suggest the authors to reduce the length by making the discussion of some parameters brief, such as SRR gap, Coptic split gap, Coptic width, and resonator materials. It feels like their influence is small.
  • Since the title indicates the resonators will be used in satellite, there is very few discussion on this (except some sentences in conclusion). I would suggest the authors to modify the title or include more description of satellite applications? How does it relate to this and what would be the benefit of this system, compared to other potential candidates?

There are also some minor things that need to be modified:

  • Some abbreviations were used without full spelling. For example, ADS, CST, DNG, ENG. However, for some terms like EMR, the authors spelled out the full name for many times.
  • In section 2, “a dielectric constant of 5.96 x 107 S/m”. Firstly, dielectric constant has no unit and S/m should be unit of (optical) conductivity. Secondly, it should be 10^7 instead of 107.
  • Figure 2 is not presented properly due to some errors when converting into pdf.
  • Figure 5, why the y-axis in d is S21 instead of “magnitude of S21”? Is there any difference? It seems like they are the same.
  • Some grammar issues. For example, in section 4, “… require more factor”, “three resonance frequency are …” There are many similar cases in the text. It would be nice if these can be revised.

Author Response

As attached.

Reviewer 3 Report

In the present paper, the authors present a modified Coptic cross shaped split ring resonator (SRR) based metamaterial that exhibits a negative permittivity and refractive index with a permeability of nearly zero. The metamaterial unit cell consists of an SRR and modified Coptic cross shaped resonator providing quadruple resonance frequency at 2.02, 6.985, 9.985 and 14.425 GHz with the magnitude of -29.45, -25.44, -19.05 and -24.45 dB, respectively.

The proposed structure therefore allow to cover more frequency bands than what was previously proposed. In general, the work is fairly well presented and the steps well described, however I find that the weak point is in the originality and novelty of the work. I think that the authors can propose this work in another journal a little less concerned about originality than Materials journal. This point therefore does not allow me to recommend the publication of this article in this journal.

Author Response

As attached.

Reviewer 4 Report

This research article describes a modified Coptic cross shaped split ring resonator (SRR) based metamaterial that exhibits a negative permittivity and refractive index with a permeability of nearly zero. The novelty of the article and its impact is clear, my minor suggestions are as follows:

  1. Authors should carefully look at the formatting of the article as Figure 2 is not visible at the moment.
  2. Authors should clarify the extent to which homogenization of the medium has been performed.
  3. Authors are having an insight into the tunability possibilities of the structure. This message should be stressed.
  4. Authors are missing some recent articles related to the investigations of the metamaterials such as Surface plasmon polaritons at the interface of two nanowire metamaterials, Metamaterial Structures and Possibility of their Application in Microwave Applicator Optimization, etc.

Author Response

As attached.

Round 2

Reviewer 1 Report

The paper is acceptable.

Reviewer 3 Report

The authors have made changes to the introduction by better explaining how their work is positioned in relation to previous work. Considering this effort, I can recommend now the publication of the article in Materials.